# Association between housing tenure and self-rated health in Japan: Findings from a nationwide cross-sectional survey

Kimiko Tomioka*, Norio Kurumatani, Keigo Saeki

Nara Prefectural Health Research Center, Nara Medical University, Kashihara, Nara, Japan

* tkimiko@naramed-u.ac.jp

## Abstract

### Background

Many studies have reported that housing tenure (HT) is associated with health, but little is known about its association in Japan. We investigated the cross-sectional association between HT and self-rated health (SRH) among Japanese adults, taking demographic characteristics and socioeconomic status (SES) into consideration.

### Methods

We used data from a nationally representative survey conducted by the Japanese Ministry of Health, Labour and Welfare (28,641 men and 31,143 women aged $\geq$20 years). HT was divided into five categories: owner-occupied, privately rented, provided housing, publically subsidized, and rented rooms. SRH was evaluated using a single-item inventory and dichotomized into poor (very poor/poor) and good (very good/good/fair). We calculated adjusted odds ratios (OR) and their 95% confidence intervals (CI) for poor SRH with logistic regression models. Covariates included demographic factors (i.e., age, gender, marital status, family size, smoking status, and chronic medical conditions) and SES factors (i.e., education, equivalent household expenditures, and occupation).

### Results

Among analyzed participants, 75.9% were owner-occupiers and 14.6% reported poor SRH. After adjustment for all covariates, compared with owner-occupiers, private renters (OR = 1.36, 95% CI = 1.26–1.47), publically subsidized renters (OR = 1.33, 95% CI = 1.19–1.48), and residents in rented rooms (OR = 1.41, 95% CI = 1.22–1.62) were more likely to report poor SRH. Stratified analyses by SES factors showed that the association between HT and poor SRH was stronger in the socially disadvantaged than in the higher socioeconomic group.

**Data Availability Statement:** Data are available from the Ministry of Health, Labour and Welfare, Japan (https://www.mhlw.go.jp/toukei/itaku/tokumei.html) for researchers who obtain approval

to use the anonymous data in accordance with Article 36 of the Statistics Law of Japan.

**Funding:** This work was supported by the Center Administrative Expense from Nara Medical University. The funder had no role in study design, data collection and analysis, decision to publish, or preparation of the manuscript.

**Competing interests:** KS reports having received research grants from YKK AP, Inc., Ushio Inc., Tokyo Electric Power Company, EnviroLife Research Institute Co., Ltd., and Sekisui Chemical Co., Ltd. This does not alter our adherence to PLOS ONE policies on sharing data and materials.

## Conclusions

Our results show a significant association between HT and SRH, independent of socio-demographic factors. HT may deserve greater attention as an indicator of socioeconomic position in Japan.

## Introduction

Social determinants that contribute to health inequality, such as educational background, economic status, and occupational class, have received a lot of attention. Housing tenure (HT) is also considered one of the important factors in determining health [1–3].

The association between HT and health has been studied in many countries, and many previous studies have reported that owner-occupiers have better health than people who live in privately rented or publically subsidized housing [4–12]. Although HT can be taken as a surrogate indicator of social class and wealth [6], whether or not the association between HT and health is independent of socioeconomic indicators may vary across countries [5,10,11].

First, HT-related factors, such as home ownership rate, household preferences, and housing administration policies, vary depending on the country [10,11,13]. According to a report by the OECD [14], in 2016 on average 69% of households across the OECD owned a dwelling, compared to 26% of households who rented a dwelling. Although owning a home is the most common form of housing, Switzerland and Germany have a majority of renters (60% in Switzerland and 55% in Germany). Regarding housing policies, most OECD countries have a system of social rental housing, but the size of the social housing stock differs from country to country: The Netherlands, Austria, Denmark, France, and the United Kingdom have a high rate of social housing stock, accounting for 15% or more of the total housing stock. In Japan, 80% of households own their dwelling, placing Japan fifth highest in the OECD ranking of ownership rates. However, Japan has a low rate of social housing stock, comprising about 5% of the total housing stock. Inadequate housing policies may increase the proportion of economically vulnerable people living in low-rent and poorly-maintained accommodation in the private rental sector. This could have adverse health effects due to financial stress and difficulties affording health care [3].

Second, Japan has among the highest proportion of older adults aged 65 and older in the world [15]. Because older people tend to spend most of their time at home and are vulnerable to barriers and problems of the home environment [16], Japanese people may be more exposed to health risks associated with their own home. Furthermore, previous studies suggest that the housing-health association can be partly explained by neighborhood environments [17,18]. A cross-national research study in China, Japan, and South Korea has reported that the association between neighborhood social environment and self-rated health is strongest in Japan [17]. Japanese culture is greatly influenced by Confucian ideals, where harmony among people is to be valued. Japanese people may tend to place emphasis on neighborly ties and therefore their health may be more affected by their neighborhood social environment. Based on these assumptions, the HT-health association may be stronger in Japan than in other countries. Therefore, we hypothesize that HT is significantly related to the health status of Japanese people, independent of other socioeconomic factors, such as education, expenditure, and occupation.

Self-rated health (SRH) is a subjective indicator of current general health, which contains diverse aspects of health including physical and mental health, well-being, and life satisfaction

[19]. Additionally, SRH, which is frequently used in international comparative statistics [20] and epidemiological studies, has been established as an independent predictor not only of mortality in general populations [21] and young adulthood [22], but also of functional decline in community-dwelling older adults [23]. Therefore, SRH is a crucial health outcome in nationally representative health surveys.

Previous studies on HT and SRH mostly focus on two categories of HT, such as home ownership and rental [4,7,10], or owner-occupiers and social renters [5]. To the best of our knowledge, no studies have examined the association between detailed types of HT and SRH in Japan. Therefore, in this study, we used anonymized data from a nationwide cross-sectional survey targeted at households and household members throughout the country and investigated the following study questions: 1) whether HT classified into five small groups is associated with SRH among the general Japanese population; 2) whether this association is independent of demographic characteristics and socioeconomic status (SES) factors; and 3) whether the relationship with HT and SRH varies depending on social demographic attributes.

## Materials and methods

### Data

The data source in this study is based on the 2010 Comprehensive Survey of Living Conditions (CSLC) conducted by the Ministry of Health, Labour and Welfare of Japan. The details of the 2010 CSLC are explained elsewhere [24]. Briefly, the CSLC covers households and their membership throughout Japan, and has been carried out annually for the purpose of collecting basic data for the promotion of national health and social welfare. A large-scale survey is implemented every 3 years. In other years, smaller-scale surveys are conducted with simplified questionnaires leaving out health-related questions such as SRH. In January 2018, when we received the CSLC data from the Ministry of Health, Labour and Welfare, the 2010 data was the latest information provided. For the 2010 CSLC, survey slips were distributed to all households in 5,510 stratified random sampling districts (289,363 households) on June 3, and collected from 229,785 households (response rate, 79.4%). We got permission to use this data for academic research in accordance with the Statistics Act, Article 36, and accepted an offer of anonymized data. Anonymized data were scrubbed of information that had the possibility of revealing personal identity; not only personal information such as name and birthdates, but also regional information such as prefectural names. In addition, anonymized data eliminated rare households such as single-male-parent households and families who had a large age difference between husband and wife, because including these households might lead to identification of individuals. After rare households were excluded, individuals from the anonymized data were randomly selected. We were provided finally with anonymized data from 93,730 people in 36,387 households; the reduced size equivalent to that of smaller-scale surveys.

### Study participants

Fig 1 displays a flowchart of study participants. Our study's aim was to examine the cross-sectional association between HT and SRH, independent of demographic factors and SES factors. People with ADL disability were excluded from our analyses because they have high potential for reverse causation; ADL disability may be a cause of some renters requiring publically subsidized housing [5]. Additionally, people in a hospital/facility and minors under twenty years of age were not required respond to questions about health status and lifestyle factors such as smoking and drinking. Therefore, we excluded 33,946 persons from our analyses because of being aged <20 (n = 16,951), being in a hospital/facility (n = 998), having received long-term

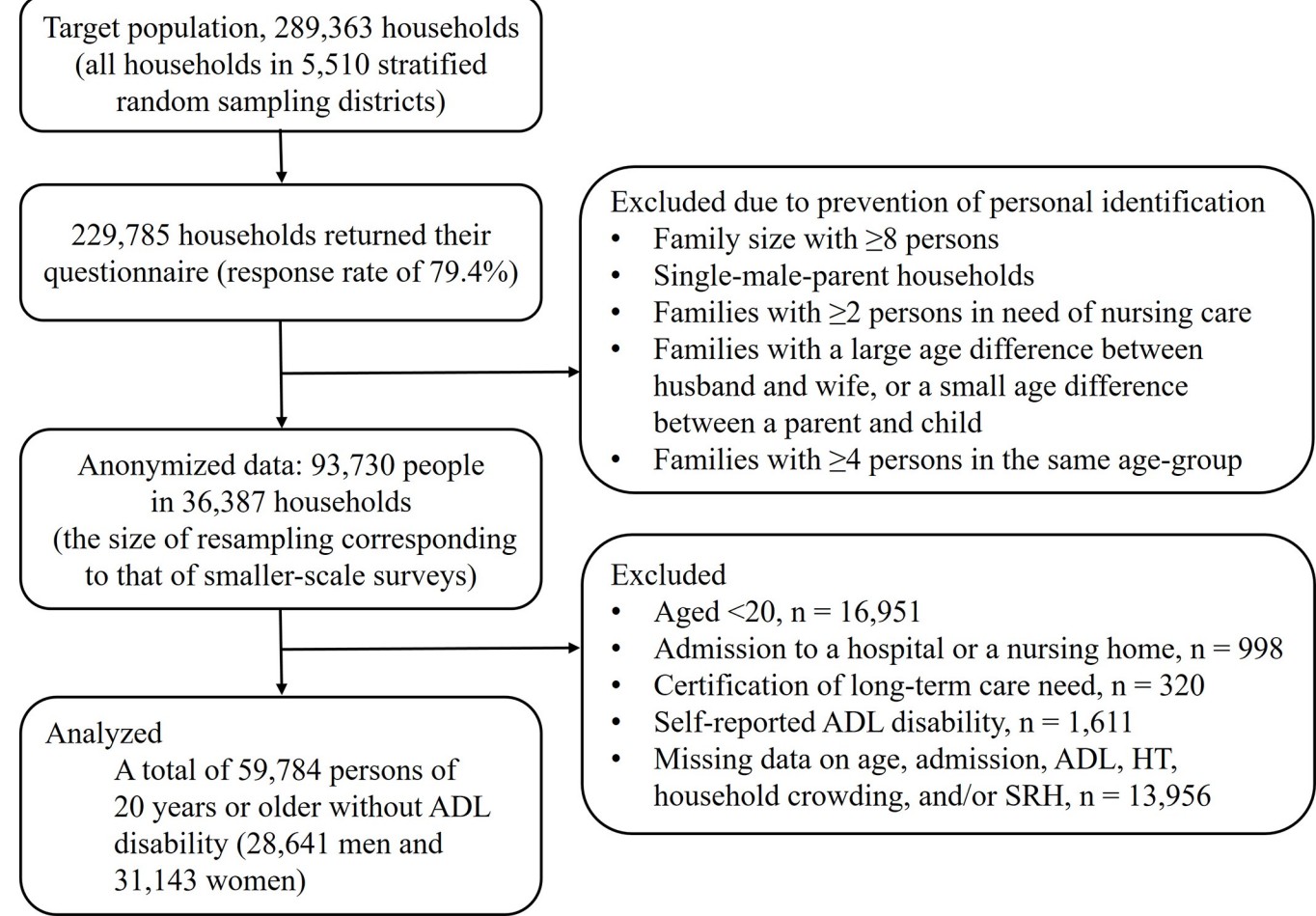

**Fig 1. Selection of study participants.** ADL, activities of daily living; HT, housing tenure; SRH, self-rated health.

care needing certification (n = 320), having a self-reported ADL disability (n = 1,611), and missing data on age, hospital admission, ADL, HT, household crowding, and/or SRH (n = 13,956). Eventually, we restricted our analyses to the data of 59,784 persons of 20 years or older (28,641 men and 31,143 women) without ADL disability whose HT, household crowding, and SRH (shown below) were available. Individuals excluded from this study due to missing information were older than those included in our analyses, but there was no gender difference between the two groups (data is provided in S1 Table).

## Measurements

**Housing tenure (HT).** The CSLC included the following question: "Which type of housing do you live in?" Participants were asked to select 1 of 5 choices: one's own house, privately rented housing, provided housing, publically subsidized housing, and rented rooms and other. According to the explanation of the terms used in the CSLC [25], provided housing is defined as company housing and housing for national public employees and local government officers, while rented rooms are defined as rented accommodation which is part of a dwelling where other households live. Based on this definition, HT was classified into five groups: owner-occupied, privately rented, provided housing, publically subsidized, and rented rooms.

**Self-rated health (SRH).** SRH has been used as an effective indicator of overall health status not only in Japan [23,26–28] but also worldwide [10,29,30]. This study assessed SRH by a single item: "How is your health in general? Is it very good, good, fair, poor, very poor?" The OECD Health Statistics [15] has recommended this as a standard form of question about perceived health status, and showed that the rate of people who report their health as good or better is very low in Japan, about 30% in 2011 compared to about 70% on average in the OECD and about 90% in the United States, New Zealand, and Canada. One reason why Japan has a low rate of people reporting to be in good health is that Japanese people have a tendency to avoid giving a direct answer and like moderation, with the result that in responses to questionnaires, there is a marked tendency to concentrate on a mid-point [31]. Therefore, although the OECD Health Statistics has considered people rating their health to good or very good as those who are in good health, Japanese epidemiological surveys have commonly adopted the definition of good health as including the middle scale of SRH [26–28]. In this study, persons whose responses were very good, good, and fair were defined as having good SRH, and poor and very poor as having poor SRH.

**Other housing information (household crowding).** The questionnaire asked all participants about the number of rooms in their dwelling excluding entrance halls and bathrooms. With reference to previous studies [4,10], household crowding is defined by the number of people in a family divided by the number of rooms. Using the median of household crowding, participants were dichotomized into low (i.e., not crowded) and high (i.e., crowded).

**Covariates.** With reference to previous studies on housing and health [4–13], the following variables were adopted as essential and/or potential covariates: demographic factors (i.e., age, gender, marital status, family size, smoking status, and chronic medical conditions), and SES factors (i.e., education, equivalent household expenditures, and occupation). Age was classified into 20–34 years, 35–49 years, 50–64 years, 65–74 years, and ≥75 years. Chronic medical conditions were defined as persons with at least one disease under treatment for hypertension, diabetes mellitus, cerebrovascular disease, heart disease, or cancer. Educational attainment (years of education) was categorized into university (≥13 years), high school (10–12 years), and junior high school (<10 years).

The CSLC asked all households about their monthly household expenditures, with the following accompanying explanation: household expenditures are defined as the total amount of money spent by all members of the household during May 2010, which includes costs of food and drink, housing, utilities, clothing, healthcare, education, entertainment, and family occasions such as marriages etc., but exclude taxes, social insurance premiums, savings, debt/mortgage repayment, and life/non-life insurance premiums other than insurance with no refund payment [25]. Equivalent household expenditures (EHE) were calculated as monthly household expenditures divided by the square root of the number of people per household [26]. Using EHE tertiles, respondents were divided into high (upper tertile, >161 Japanese thousand yen), moderate (middle tertile, 106–161 Japanese thousand yen), and low (lower tertile, <106 Japanese thousand yen).

Regarding occupation, respondents were asked about their working status in the last month. Additionally, persons with a job were asked about their occupation based on the Japan Standard Occupational Classification [32], which is compatible with the International Standard Classification of Occupations. According to these two questions, participants were grouped into four classes; upper non-manual (i.e., managers, professionals, and technicians), lower non-manual (i.e., clerical, sales, and services workers), manual (i.e., manufacturing, transport, machine, construction, mining, protective service, agricultural, forestry, fishery, carrying, cleaning, and packing workers), and non-working [33].

In considering missing values in response to questions, we used missing category of covariates in the multivariable statistical models [34]. Characteristics of study participants, including the number of missing values, are provided in S2 Table.

## Statistical analysis

Multiple logistic regression analysis (by the forced entry method) was carried out using 'poor SRH' or 'good SRH' as a dependent variable. Independent variables were the five types of HT (i.e., owner-occupied housing, privately rented housing, provided housing, publically subsidized housing, and rented rooms), with owner-occupied housing as the reference group. The results were shown as an odds ratio (OR) with a 95% confidence interval (CI) for poor SRH. Model 1 was adjusted for age and gender. Model 2 was adjusted for all demographic factors (i.e., age, gender, marital status, family size, smoking status, and chronic medical conditions). Model 3 was adjusted for SES factors (i.e., education, EHE, and occupation), in addition to adjustment for the variables in Model 2.

There was a high correlation between HT and household crowding (e.g., home renters tend to live in more crowded homes than owner-occupiers). To avoid the issue of multicollinearity, we abandoned the use of household crowding as a covariate and considered the association between household crowding and poor SRH according to type of HT.

All statistical analyses were performed using IBM SPSS Statistics version 24.0 (IBM Corporation, Armonk, NY, USA), and the significance threshold was set at $P < 0.05$ (two-sided).

## Ethics

In this study, we received approval of use for academic purposes from the Japanese Ministry of Health, Labour and Welfare and were provided data without any information that would identify individuals.

## Results

### Participant characteristics

Of the 59,784 participants, 24.1% were older adults aged 65 years or older, 47.9% were men, 75.9% were owner occupiers, and 14.6% reported poor SRH. Table 1 shows the characteristics of study participants by HT status. Owner-occupiers were more likely to be aged 65 years and older and have chronic medical conditions, and less likely to be current smokers and live in crowded houses; and private renters had the second highest tobacco use; because a large number of people living in provided housing live apart from their family at the new workplace, they had the highest percentage of the married, males, and upper non-manual workers, the second highest percentage of persons living alone, and the lowest percentage of persons aged 65 or older and those who had a low level of education. Publically subsidized renters tended to have junior high school education, live in crowded homes, and report poor SRH; and those living in rented rooms were more likely to be current smokers and have a low EHE.

### Cross-sectional association between HT and SRH

The ORs for poor SRH associated with HT are presented in Table 2. Publically subsidized renters had a significantly higher prevalence of poor SRH. This significant relationship persisted after adjustment for demographic factors and SES factors (adjusted OR 1.33; 95% CI, 1.19–1.48 in Model 3). Persons living in provided housing were less likely to have poor SHR than owner-occupiers. However, after adjustment for age and gender, this association did not remain significant: adjusted OR (95% CI) was 1.11 (0.94–1.32) in model 1 and 1.12 (0.94–1.34)

**Table 1. Characteristics of study participants by housing tenure status.**

| | | | All participants | Housing tenure status | | | | | |
|---|---|---|---|---|---|---|---|---|---|
| | | | | Owner-occupied | Privately rented | Provided housing | Publically subsidized | Rented rooms | P-value[a] |
| | | | (n = 59,784) | (n = 45,354) | (n = 8,423) | (n = 1,461) | (n = 2,733) | (n = 1,813) | |
| Age: | 20–44 years | % | 39.3 | 32.8 | 63.9 | 70.0 | 44.3 | 56.9 | <0.001 |
| | 45–64 years | % | 36.6 | 39.3 | 26.1 | 27.3 | 32.3 | 30.9 | <0.001 |
| | ≥65 years | % | 24.1 | 27.9 | 9.9 | 2.7 | 23.3 | 12.1 | <0.001 |
| Gender: men | | % | 47.9 | 47.4 | 50.3 | 55.9 | 44.7 | 48.9 | <0.001 |
| Marital status: married | | % | 67.4 | 69.9 | 58.7 | 72.4 | 59.5 | 52.1 | <0.001 |
| Family size: one (living alone) | | % | 10.7 | 5.9 | 27.2 | 29.6 | 15.3 | 30.8 | <0.001 |
| Smoking: current smokers | | % | 23.6 | 21.4 | 31.7 | 23.8 | 29.3 | 31.4 | <0.001 |
| Medical conditions: present[b] | | % | 41.3 | 44.4 | 29.8 | 28.7 | 39.2 | 30.2 | <0.001 |
| Education: junior high school | | % | 13.7 | 14.4 | 10.4 | 4.4 | 17.7 | 11.9 | <0.001 |
| Household expenditures[c] | | mean ± SD | 14.7 ± 8.2 | 14.8 ± 8.3 | 14.9 ± 7.7 | 14.5 ± 8.4 | 14.1 ± 8.2 | 12.7 ± 7.6 | <0.001 |
| Occupation: upper non-manual | | % | 19.4 | 18.7 | 22.5 | 35.2 | 13.9 | 18.9 | <0.001 |
| Household crowding[d] | | mean ± SD | 0.66 ± 0.33 | 0.60 ± 0.28 | 0.86 ± 0.39 | 0.81 ± 0.33 | 0.87 ± 0.39 | 0.79 ± 0.42 | <0.001 |
| Self-rated health: poor | | % | 14.6 | 14.6 | 14.4 | 11.1 | 17.3 | 15.4 | <0.001 |

[a]Differences between the five groups were analyzed using the Chi-squared test for categorical variables and the analysis of variance for continuous variables.

[b]Persons being treated for at least one of hypertension, diabetes mellitus, cerebrovascular disease, heart disease, and cancer.

[c]Monthly equivalent household expenditures (unit: Japanese one-thousand yen)

[d]The number of people in a family divided by the number of rooms. A higher value indicates more crowded.

in model 3. In contrast, in the crude model, private renters and persons inhabiting rented rooms showed no association with poor SRH relative to owner-occupiers. After adjustment for age and gender, private renters and residents in rented rooms were more likely to have poor SRH than owner-occupiers. These significant associations were unchanged after adjustment for SES factors as well as demographic factors (Model 3): adjusted OR (95% CI) was 1.36 (1.26–1.47) in private renters and 1.41 (1.22–1.62) in residents in rented rooms. Given that nine out of every ten people in this study had more than one member in their household, there was a possibility that we had the violation of the independence assumption in the association between HT and SRH. To address this concern, we conducted subset analyses limited to the head of the household (n = 27,849). The results did not change significantly with the subset analyses: in Model 3, adjusted OR (95% CI) was 1.46 (1.32–1.62) in private renters, 1.43 (1.24–1.65) in publically subsidized renters, and 1.35 (1.12–1.61) in residents in rented rooms, compared to owner-occupiers.

## Consideration of household crowding

The association between household crowding and SRH by HT is shown in **Table 3**. In the crude model, a significantly lower OR for household crowding of having poor SRH compared with non-crowded homes was found in owner-occupiers, private renters, and publically

**Table 2. Odd ratios for poor self-rated health associated with housing tenure.**

| Housing tenure | n | Crude model | | Model 1[a] | | Model 2[b] | | Model 3[c] | |
|---|---|---|---|---|---|---|---|---|---|
| | | OR (95% CI) | P-value | OR (95% CI) | P-value | OR (95% CI) | P-value | OR (95% CI) | P-value |
| All study participants (n = 59,784) | | | | | | | | | |
| Owner-occupied | 45,354 | 1.00 | | 1.00 | | 1.00 | | 1.00 | |
| Privately rented | 8,423 | 0.98 (0.92–1.05) | 0.619 | 1.35 (1.26–1.45) | <0.001 | 1.38 (1.28–1.49) | <0.001 | 1.36 (1.26–1.47) | <0.001 |
| Provided housing | 1,461 | 0.73 (0.62–0.86) | <0.001 | 1.11 (0.94–1.32) | 0.214 | 1.12 (0.94–1.33) | 0.206 | 1.12 (0.94–1.34) | 0.206 |
| Publically subsidized | 2,733 | 1.23 (1.11–1.36) | <0.001 | 1.37 (1.23–1.52) | <0.001 | 1.37 (1.23–1.53) | <0.001 | 1.33 (1.19–1.48) | <0.001 |
| Rented rooms | 1,813 | 1.06 (0.93–1.21) | 0.357 | 1.38 (1.21–1.58) | <0.001 | 1.44 (1.26–1.66) | <0.001 | 1.41 (1.22–1.62) | <0.001 |
| Limited to the householders (n = 27,849) | | | | | | | | | |
| Owner-occupied | 19,045 | 1.00 | | 1.00 | | 1.00 | | 1.00 | |
| Privately rented | 5,241 | 0.94 (0.86–1.02) | 0.127 | 1.43 (1.30–1.57) | <0.001 | 1.47 (1.33–1.63) | <0.001 | 1.46 (1.32–1.62) | <0.001 |
| Provided housing | 928 | 0.60 (0.48–0.74) | <0.001 | 1.09 (0.87–1.36) | 0.473 | 1.17 (0.93–1.47) | 0.191 | 1.24 (0.98–1.57) | 0.069 |
| Publically subsidized | 1,481 | 1.31 (1.15–1.50) | <0.001 | 1.46 (1.27–1.67) | <0.001 | 1.48 (1.29–1.71) | <0.001 | 1.43 (1.24–1.65) | <0.001 |
| Rented rooms | 1,154 | 0.94 (0.80–1.11) | 0.478 | 1.34 (1.13–1.60) | 0.001 | 1.39 (1.16–1.66) | <0.001 | 1.35 (1.12–1.61) | 0.001 |

CI, confidence interval; OR, odds ratio.

[a]Adjusted for age and gender.

[b]In addition to Model 1, marital status, family size, smoking status, and chronic medical conditions were included.

[c]In addition to Model 2, socioeconomic status factors (i.e., education, equivalent household expenditures, and occupation) were included.

subsidized renters. After adjustment for age and gender (Model 1), significant associations in private renters and publically subsidized renters disappeared. After adjustment for all demographic factors (Model 2), owner-occupiers had no association between household crowding and SRH. In contrast, for those living in rented rooms, household crowding was not associated

**Table 3. Association between household crowding and self-rated health by housing tenure.**

| | n | Crude model | | Model 1[a] | | Model 2[b] | | Model 3[c] | |
|---|---|---|---|---|---|---|---|---|---|
| | | OR (95% CI) | P-value | OR (95% CI) | P-value | OR (95% CI) | P-value | OR (95% CI) | P-value |
| Owner-occupied housing (n = 45,354) | | | | | | | | | |
| Not crowded | 27,599 | 1.00 | | 1.00 | | 1.00 | | 1.00 | |
| Crowded | 17,755 | 0.70 (0.67–0.74) | <0.001 | 0.94 (0.88–0.99) | 0.029 | 1.01 (0.94–1.08) | 0.847 | 1.00 (0.93–1.07) | 0.916 |
| Privately rented housing (n = 8,423) | | | | | | | | | |
| Not crowded | 2,263 | 1.00 | | 1.00 | | 1.00 | | 1.00 | |
| Crowded | 6,160 | 0.76 (0.67–0.87) | <0.001 | 0.96 (0.84–1.11) | 0.590 | 1.06 (0.90–1.25) | 0.472 | 1.04 (0.88–1.23) | 0.647 |
| Provided housing (n = 1,461) | | | | | | | | | |
| Not crowded | 422 | 1.00 | | 1.00 | | 1.00 | | 1.00 | |
| Crowded | 1,039 | 0.87 (0.61–1.24) | 0.439 | 1.05 (0.72–1.52) | 0.813 | 1.08 (0.69–1.70) | 0.729 | 1.05 (0.67–1.67) | 0.821 |
| Publically subsidized housing (n = 2,733) | | | | | | | | | |
| Not crowded | 706 | 1.00 | | 1.00 | | 1.00 | | 1.00 | |
| Crowded | 2,027 | 0.63 (0.51–0.77) | <0.001 | 0.88 (0.70–1.11) | 0.279 | 0.99 (0.72–1.36) | 0.951 | 0.98 (0.71–1.35) | 0.880 |
| Rented rooms (n = 1,813) | | | | | | | | | |
| Not crowded | 704 | 1.00 | | 1.00 | | 1.00 | | 1.00 | |
| Crowded | 1,109 | 0.94 (0.72–1.22) | 0.625 | 1.30 (0.98–1.73) | 0.065 | 1.40 (1.04–1.90) | 0.027 | 1.43 (1.05–1.95) | 0.024 |

CI, confidence interval; OR, odds ratio.

[a]Adjusted for age and gender.

[b]In addition to Model 1, marital status, family size, smoking status, and chronic medical conditions were included.

[c]In addition to Model 2, education, equivalent household expenditures, and occupation were included.

with SRH in crude model. After adjustment for demographic factors (Model 2), persons living in crowded homes were more likely to have poor SRH than persons without crowded homes. These significant associations remained after additional adjustment for SES factors (Model 3): adjusted OR (95% CI) was 1.43 (1.05–1.95) in persons inhabiting crowded rented rooms, compared to those living in non-crowded rented rooms.

### Additional stratified analyses

We conducted further stratified analyses by age, gender, chronic medical conditions, and SES factors because these variables can affect the association between HT and SRH [4–7,26,28,35] (results are shown in **S3 and S4 Tables**). HT had a greater impact on SHR for persons aged 45 or older than for those aged 20–44 and for women than for men. Chronic medical conditions did not affect the association between HT and SRH. Regarding education, the lower level the participants had, the more their SRH was affected by HT. EHE had only a slight influence on the association between HT and SRH. For occupation, among upper non-manual workers, HT had no effect on SRH. The association between HT and poor SRH was the strongest in non-working people. Generally, these results indicated that the HT-SRH association was stronger in the socially disadvantaged than in the higher socioeconomic group.

## Discussion

Our study found that publically subsidized renters tended to have significantly poorer SRH than owner-occupiers, independent of demographic and SES factors. This result supports our hypothesis as well as agrees with the results of previous studies on HT and health in non-Japanese countries [4–7,10,12]. Additionally, we have revealed that not only residents in publically subsidized housing but also those in other types of HT had a significantly worse SRH than owner-occupiers. Our results based on stratified analyses showed that HT status had a greater association with poor SRH among people with weaker social positions than those with a higher level of socio-demographic status. Our findings are consistent with those of previous studies [1,11,35] which suggest that housing environment exerts a greater impact on the health of vulnerable groups such as ethnic minorities, older adults, and the unemployed than the socially advantaged.

The association between HT and SRH may be explained by three mechanisms: housing circumstances, health behaviors, and the neighborhood. First, previous studies suggest that the housing environment of vulnerable groups tends to be crowded due to a large number of people living in a limited area, unsanitary living conditions due to inadequate garbage and sewage treatment, bad air with harmful chemical substances and poor ventilation, and poor hygro-thermal conditions such as warmth and humidity [3,36]. Crowding and poor air quality can cause respiratory disease and communicable diseases, a polluted water supply can spread waterborne diseases, and cold homes can elevate the risk for cardiovascular diseases and poor mental health [36,37]. In Japan, because universal access to a clean water supply and an effective sanitation system has improved the level of public hygiene, unsanitary living conditions may not be an issue nowadays. However, because Japan has a high death toll during the winter months [38], excess cold in the home is an ongoing problem in Japan: People with financial difficulties may not be able to afford adequate heating, with consequent negative impact on their health. In this study, among persons living in rented rooms, household crowding was associated with SRH. Our results suggest that people living in rented rooms may be exposed to more hazards in the home environment than people living in other types of HT. For example, persons inhabiting crowded rented rooms seem likely to face the risk of intruders and excess

noise. These conditions together with crowding in housing are considered as psychological hazards relating to housing, and could lead to increased risk of illness [37].

Second, previous studies have reported that the social housing group tend to have a lower level of physical activity and a higher prevalence of obesity than those in other housing sectors [39] and that house owners are more likely to have regular medical checkups [40] and to be non-smokers [11] than persons without house ownership; they indicate that the positive health effect of house ownership may be the result of healthy lifestyle and active healthy behaviors among owner-occupiers. These findings from previous studies agree with our results showing that, among the five types of HT, house owners have the lowest rate of current smokers.

Third, the neighborhood environment of dwellings in the publically rented sector is often characterized by poor access to recreational facilities, shops, public transport, social support networks, and health services [6,39]. A neighborhood environment that discourages physical activity and health care may affect health behaviors and the health of residents [3]. On the other hand, research has shown that people living in a neighborhood with more sports/recreational facilities, more walkable green spaces, and more people who practice healthy lifestyles such as choosing healthy foods and doing active physical exercise are more likely to report being in good health [41–44]. The communities with a high homeownership rate are considered to have a healthy neighborhood [11], which promotes the health of house owners.

Our study has several strengths. First, this study is based on nationally representative data. Therefore, we have the advantage of generalization and have been able to examine the SRH-association based on five categories of HT. Second, the CSLC collects a large number of sociodemographic indicators such as marital status, chronic medical conditions, education, and occupation. The use of many covariates in this study makes the validity of the results a strong point.

Our study has several limitations. First, we cannot omit the effect of reverse causation due to cross-sectional design. In particular, we should consider the fact that health problems can trigger unemployment and loss of stable employment [5,45]. As a result, people may end up in poverty and need to sell their houses. In this case, poor health status is not attributed to housing environment. However, it is highly likely that residents in privately rented residences, publically subsidized housing, and rented rooms who relocate due to health issues will have poor SRH. Second, our results are based on self-assessment. Therefore, we have the possibility of problems with common method variance and an overestimation between HT and SRH [46]. Future studies are needed to confirm the association between HT and health using an objective assessment of health based on a prospective study design. Third, anonymized data eliminated the rare households in order to prevent identification of individuals, and our study failed to include persons with missing data. The former ruled out social minorities and the latter resulted in selective loss of older people. That is, persons omitted from our study are vulnerable groups, and potentially have poor health and poor housing. This bias might lead to an underestimation of the association between HT and SRH in this study.

## Conclusions

This study has revealed a significant association between HT and SRH in Japan, independent of SES factors and demographic characteristics. The Japanese government has adopted reduction of health disparities as one of the main target objectives for the national health promotion plan, Health Japan 21 (the second term) [47]. For a better understanding of disparities in health status among the Japanese, our findings suggest that HT is an important factor that deserves notice. From a policy perspective, policymakers need to pay more attention to HT as a social determinant that contributes to health inequality in Japan.

## Supporting information

**S1 Table. Basic attributes of individuals included in this study and those excluded from analyses.**
(DOCX)

**S2 Table. Characteristics of study participants.**
(DOCX)

**S3 Table. Adjusted odds ratios for poor self-rated health based on stratified analyses by age, gender, and chronic medical conditions.**
(DOCX)

**S4 Table. Adjusted odds ratios for poor self-rated health based on stratified analyses by socioeconomic status factors.**
(DOCX)

## Acknowledgments

We wish to thank the participants of this study and support staff who make the study possible. We also thank Dr. Heather Hill for her English language editing.

## Author Contributions

**Conceptualization:** Kimiko Tomioka, Norio Kurumatani.

**Data curation:** Kimiko Tomioka, Keigo Saeki.

**Formal analysis:** Kimiko Tomioka.

**Funding acquisition:** Kimiko Tomioka, Norio Kurumatani.

**Methodology:** Kimiko Tomioka, Keigo Saeki.

**Project administration:** Norio Kurumatani.

**Supervision:** Norio Kurumatani, Keigo Saeki.

**Validation:** Keigo Saeki.

**Visualization:** Kimiko Tomioka.

**Writing – original draft:** Kimiko Tomioka.

**Writing – review & editing:** Norio Kurumatani, Keigo Saeki.

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
