## [Decision Letter · Decision Letter 0]

6 Sep 2019

PONE-D-19-16134

Association between Housing Tenure and Self-Rated Health in Japan: Findings from a Nationwide Cross-Sectional Survey

PLOS ONE

Dear Dr Tomioka,

Thank you for submitting your manuscript to PLOS ONE. After careful consideration, we feel that it has merit but does not fully meet PLOS ONE’s publication criteria as it currently stands. Therefore, we invite you to submit a revised version of the manuscript that addresses the points raised during the review process.

We would appreciate receiving your revised manuscript by Oct 21 2019 11:59PM. To enhance the reproducibility of your results, we recommend that if applicable you deposit your laboratory protocols in protocols.io, where a protocol can be assigned its own identifier (DOI) such that it can be cited independently in the future. For instructions see: http://journals.plos.org/plosone/s/submission-guidelines#loc-laboratory-protocols

We look forward to receiving your revised manuscript.

Kind regards,

Sungwoo Lim, DrPH

Academic Editor

PLOS ONE

Journal Requirements:

Reviewers' comments:

Reviewer's Responses to Questions

**Comments to the Author**

1. Is the manuscript technically sound, and do the data support the conclusions?

Reviewer #1: Yes

Reviewer #2: Yes

2. Has the statistical analysis been performed appropriately and rigorously? 

Reviewer #1: Yes

Reviewer #2: Yes

3. Have the authors made all data underlying the findings in their manuscript fully available?

Reviewer #1: Yes

Reviewer #2: Yes

4. Is the manuscript presented in an intelligible fashion and written in standard English?

Reviewer #1: Yes

Reviewer #2: Yes

5. Review Comments to the Author

Reviewer #1: This is a well written paper describing research on housing and health in underresearched context of Japan.

As authors note, the surrounding neighbourhood may be part of the explanation on why housing tenure might matter, so it would be useful for readers if authors could include research on neighbourhoods and health in Japan. There isn't much but some to look at include

Lui et al https://doi.org/10.1371/journal.pone.0204910

loo et al https://doi.org/10.1080/24694452.2016.1271306

Reviewer #2: Thank you for the opportunity to review this paper. While this study is not novel, I do believe it makes a worthy contribution to the literature, as it provides evidence that the housing tenure-health relationship holds in the Japanese context. The methodology is sound and so are the conclusions that flow from the results. I have two main suggestions for improvement.

Firstly, the paper would be strengthened by providing more background information about the Japanese context (i.e., provide rationale for why it is important to examine this relationship in Japan). Is there any reason to believe the housing tenure-health relationship would not hold in Japan?

Secondly, the paper is concise, well-organized, and flows. However, while the meaning is clear, there are quite a few instances where the English wording/phrasing/grammar choices are unconventional (e.g., “for the purpose of gathering fundamental materials”. The paper would be improved by being edited for style, grammar, and word usage.

Below are my detailed comments about each section of the manuscript.

Abstract

- The abstract is clearly written.

Introduction

- There are a couple of instances where the number for the reference isn’t in brackets (e.g., Introduction - 2nd paragraph, 4th line, 6 should be in brackets)

- I recommend adding a sentence or two to explain how HT-related factors (e.g., home ownership rate and housing administration policies) may affect the HT-health relationship.

- I recommend that after this sentence “To the best of our knowledge, no studies have examined the association between detailed types of HT and health in Japan”, an explanation of how the Japanese context is unique and why this is worthy avenue of study is added. Why wouldn’t the HT-health relationship hold in Japan? Please include more rationale for this study.

- Please write the study questions as questions, as they are not written as questions in their current form.

- I recommend explaining/clarifying the difference between questions 2 and 3.

Methods

- Please address the following questions:

o Why was the 2010 survey used and not a more recent one? It was mentioned that the CSLC is an annual survey.

o Why were the researchers limited to a sample of 36,387 households when data was collected from 229,785 households (15% of the original size)?

o Was more than one household member included in the sample? If so, did the authors control for potential clustering (i.e., violation of the independence assumption)?

o What was the rationale for dichotomizing SRH?

o Is the dependent variable SRH (not poor SRH and good SRH was the reference category)?

- Suggestions to increase clarity:

o Restate the independent variable as being HT, which has five categories (owner-occupier was the reference category)

o Change ‘smoking habit’ to ‘smoking status’

o Adding a flow chart showing how the cohort was created would be helpful (and would make the text in this section more concise). Suggestion to include the percentage of people excluded for each reason in addition to the frequencies.

o I had to do a little background research to understand why Poisson regression was used and not logistic regression. It might be helpful to include a one sentence explanation on why Poisson regression was performed.

- Thanks for including the S1 Table. In the limitation section, please address any bias that may have resulted by excluding a group of people who were older than those included in the study.

Results

- It seems a bit odd that that those who were living in provided housing had the highest percentage married (more than 70%) and the highest percentage living alone (almost 30%). I recommend addressing this seemingly conflicting finding.

- The statistical test results are missing from Table 1. It was noted below the table that chi-square and ANOVA tests were performed.

Discussion

- This study did not include any minority groups; suggestion to drop mention of this as a vulnerable group.

- The authors suggest that the housing environment may contribute to poor health. Is unsanitary conditions, inadequate garbage disposal and sewage treatment, poor ventilation, etc. a problem in Japan?

- In the Japanese context, why might home owners engage in healthier behaviors than other HT groups? In Japan, is social housing found in neighbourhoods with lower social capital?

- From a policy perspective, how are these results useful?

General Questions:

- Does the journal have a policy about the use of the term ‘subject’ versus ‘participant’?

- The authors used the term ‘gender’, not ‘sex’. Which word should be used in this context?

6. PLOS authors have the option to publish the peer review history of their article (what does this mean?). If published, this will include your full peer review and any attached files.

Reviewer #1: No

Reviewer #2: No

---

## [Author Response · Author response to Decision Letter 0]

11 Oct 2019

To the Editor:

We appreciate the opportunity to address the Reviewers’ comments and revise our manuscript. Below, please find item-by-item responses to the Reviewers’ comments, which are included verbatim. We apologize for the mistake of excluding 3,557 participants whose housing tenure, household crowding, and self-rated health were available. The cause of this mistake was the following: At the initial attempt, health variables included two indicators (i.e., self-rated health and psychological distress). Participants who were excluded from the original analysis were people with missing data on psychological distress. These participants were included in the revised analysis. As such, the number of analyzed participants has increased by 3,557 to a total of 59,784, although the analysis with 59,784 participants yielded similar results. Our responses are in blue font below, while the revisions in the main text are in red.

 

Response to Comments from Reviewer #1:

Reviewer #1: This is a well written paper describing research on housing and health in underresearched context of Japan.

Response:

We thank the Reviewer #1 for his/her favorable evaluation of and constructive comments on our manuscript. We have revised the manuscript accordingly, and explain the revised parts one by one as follows. We apologize for the mistake of excluding 3,557 participants whose housing tenure, household crowding, and self-rated health were available. The cause of this mistake was the following: At the initial attempt, health variables included two indicators (i.e., self-rated health and psychological distress). Participants who were excluded from the original analysis were people with missing data on psychological distress. These participants were included in the revised analysis. As such, the number of analyzed participants has increased by 3,557 to a total of 59,784, although the analysis with 59,784 participants yielded similar results. Our responses are in blue font below, while the revisions in the main text are in red.

Comment 1. As authors note, the surrounding neighbourhood may be part of the explanation on why housing tenure might matter, so it would be useful for readers if authors could include research on neighbourhoods and health in Japan. There isn't much but some to look at include

Liu et al https://doi.org/10.1371/journal.pone.0204910

Loo et al https://doi.org/10.1080/24694452.2016.1271306

Response: 

Thank you very much for introducing us to useful recent articles about neighborhoods and health in Japan. As described in Liu et al. (2018), Japan has the greater association between neighborhood social environment and self-rated health than South Korea and China. In the revised draft, we have added new references including the two provided by you, and have rewritten the Introduction and Discussion sections as follows:

Revised parts: 

Page 5-6, Line 55-68.

Second, Japan has among the highest proportion of older adults aged 65 and older in the world [15]. Because older people tend to spend most of their time at home and are vulnerable to barriers and problems of the home environment [16], Japanese people may be more exposed to health risks associated with their own home. Furthermore, previous studies suggest that the housing-health association can be partly explained by neighborhood environments [17,18]. A cross-national research study in China, Japan, and South Korea has reported that the association between neighborhood social environment and self-rated health is strongest in Japan [17]. Japanese culture is greatly influenced by Confucian ideals, where harmony among people is to be valued. Japanese people may tend to place emphasis on neighborly ties and therefore their health may be more affected by their neighborhood social environment. Based on these assumptions, the HT-health association may be stronger in Japan than in other countries. Therefore, we hypothesize that HT is significantly related to the health status of Japanese people, independent of other socioeconomic factors, such as education, expenditure, and occupation.

Page 26, Line 334-343.

Third, the neighborhood environment of dwellings in the publically rented sector is often characterized by poor access to recreational facilities, shops, public transport, social support networks, and health services [6,39]. A neighborhood environment that discourages physical activity and health care may affect health behaviors and the health of residents [3]. On the other hand, research has shown that people living in a neighborhood with more sports/recreational facilities, more walkable green spaces, and more people who practice healthy lifestyles such as choosing healthy foods and doing active physical exercise are more likely to report being in good health [41-44]. The communities with a high homeownership rate are considered to have a healthy neighborhood [11], which promotes the health of house owners.

 

Response to Comments from Reviewer #2:

Reviewer #2: Thank you for the opportunity to review this paper. While this study is not novel, I do believe it makes a worthy contribution to the literature, as it provides evidence that the housing tenure-health relationship holds in the Japanese context. The methodology is sound and so are the conclusions that flow from the results. I have two main suggestions for improvement.

Response:

We thank the Reviewer #2 for his/her favorable evaluation of and constructive comments on our manuscript. We have revised the manuscript accordingly, and explain the revised parts one by one as follows. We apologize for the mistake of excluding 3,557 participants whose housing tenure, household crowding, and self-rated health were available. The cause of this mistake was the following: At the initial attempt, health variables included two indicators (i.e., self-rated health and psychological distress). Participants who were excluded from the original analysis were people with missing data on psychological distress. These participants were included in the revised analysis. As such, the number of analyzed participants has increased by 3,557 to a total of 59,784, although the analysis with 59,784 participants yielded similar results. Our responses are in blue font below, while the revisions in the main text are in red.

Comment 1. Firstly, the paper would be strengthened by providing more background information about the Japanese context (i.e., provide rationale for why it is important to examine this relationship in Japan). Is there any reason to believe the housing tenure-health relationship would not hold in Japan?

Response: 

Thank you very much for providing us with a valuable comment. In response to your comment, we have given additional background information on the Japanese context in the Introduction section and revised the Discussion section as follows:

Revised parts: 

Page 4-6, Line 39-68 (Introduction).

First, HT-related factors, such as home ownership rate, household preferences, and housing administration policies, vary depending on the country [10,11,13]. According to a report by the OECD [14], in 2016 on average 69% of households across the OECD owned a dwelling, compared to 26% of households who rented a dwelling. Although owning a home is the most common form of housing, Switzerland and Germany have a majority of renters (60% in Switzerland and 55% in Germany). Regarding housing policies, most OECD countries have a system of social rental housing, but the size of the social housing stock differs from country to country: The Netherlands, Austria, Denmark, France, and the United Kingdom have a high rate of social housing stock, accounting for 15% or more of the total housing stock. In Japan, 80% of households own their dwelling, placing Japan fifth highest in the OECD ranking of ownership rates. However, Japan has a low rate of social housing stock, comprising about 5% of the total housing stock. Inadequate housing policies may increase the proportion of economically vulnerable people living in low-rent and poorly-maintained accommodation in the private rental sector. This could have adverse health effects due to financial stress and difficulties affording health care [3].

Second, Japan has among the highest proportion of older adults aged 65 and older in the world [15]. Because older people tend to spend most of their time at home and are vulnerable to barriers and problems of the home environment [16], Japanese people may be more exposed to health risks associated with their own home. Furthermore, previous studies suggest that the housing-health association can be partly explained by neighborhood environments [17,18]. A cross-national research study in China, Japan, and South Korea has reported that the association between neighborhood social environment and self-rated health is strongest in Japan [17]. Japanese culture is greatly influenced by Confucian ideals, where harmony among people is to be valued. Japanese people may tend to place emphasis on neighborly ties and therefore their health may be more affected by their neighborhood social environment. Based on these assumptions, the HT-health association may be stronger in Japan than in other countries. Therefore, we hypothesize that HT is significantly related to the health status of Japanese people, independent of other socioeconomic factors, such as education, expenditure, and occupation.

Page 24, Line 295-305 (Discussion)

Our study found that publically subsidized renters tended to have significantly poorer SRH than owner-occupiers, independent of demographic and SES factors. This result supports our hypothesis as well as agrees with the results of previous studies on HT and health in non-Japanese countries [4-7,10,12]. Additionally, we have revealed that not only residents in publically subsidized housing but also those in other types of HT had a significantly worse SRH than owner-occupiers. Our results based on stratified analyses showed that HT status had a greater association with poor SRH among people with weaker social positions than those with a higher level of socio-demographic status. Our findings are consistent with those of previous studies [1,11,35] which suggest that housing environment exerts a greater impact on the health of vulnerable groups such as ethnic minorities, older adults, and the unemployed than the socially advantaged.

Comment 2. Secondly, the paper is concise, well-organized, and flows. However, while the meaning is clear, there are quite a few instances where the English wording/phrasing/grammar choices are unconventional (e.g., “for the purpose of gathering fundamental materials”. The paper would be improved by being edited for style, grammar, and word usage.

Response:

In accordance with your comment, this paper has been reviewed again by an experienced editor whose first language is English. Regarding “for the purpose of gathering fundamental materials”, we have amended this to “for the purpose of collecting basic data”.

Comment 3. Introduction- There are a couple of instances where the number for the reference isn’t in brackets (e.g., Introduction - 2nd paragraph, 4th line, 6 should be in brackets)

Response:

These were our mistake, and have now been corrected.

Revised parts: 

Page 4, Line 35-38.

Although HT can be taken as a surrogate indicator of social class and wealth [6], whether or not the association between HT and health is independent of socioeconomic indicators may vary across countries [5,10,11]. 

Page 6, Line 71-74.

Additionally, SRH, which is frequently used in international comparative statistics [20] and epidemiological studies, has been established as an independent predictor not only of mortality in general populations [21] and young adulthood [22], but also of functional decline in community-dwelling older adults [23].

Comment 4. Introduction

- I recommend adding a sentence or two to explain how HT-related factors (e.g., home ownership rate and housing administration policies) may affect the HT-health relationship.

- I recommend that after this sentence “To the best of our knowledge, no studies have examined the association between detailed types of HT and health in Japan”, an explanation of how the Japanese context is unique and why this is worthy avenue of study is added. Why wouldn’t the HT-health relationship hold in Japan? Please include more rationale for this study.

- Please write the study questions as questions, as they are not written as questions in their current form.

- I recommend explaining/clarifying the difference between questions 2 and 3.

Response:

As you pointed out, the first version of our manuscript lacked any description of the rationale for this study and the study questions. To deal with these concerns which you noted, we have conducted literature searches and found some studies on the HT-health relationship and cross-national comparisons. Therefore, in the second version of our draft, we have revised the Introduction as follows:

Revised parts: 

Page 4-6, Line 39-68. 

First, HT-related factors, such as home ownership rate, household preferences, and housing administration policies, vary depending on the country [10,11,13]. According to a report by the OECD [14], in 2016 on average 69% of households across the OECD owned a dwelling, compared to 26% of households who rented a dwelling. Although owning a home is the most common form of housing, Switzerland and Germany have a majority of renters (60% in Switzerland and 55% in Germany). Regarding housing policies, most OECD countries have a system of social rental housing, but the size of the social housing stock differs from country to country: The Netherlands, Austria, Denmark, France, and the United Kingdom have a high rate of social housing stock, accounting for 15% or more of the total housing stock. In Japan, 80% of households own their dwelling, placing Japan fifth highest in the OECD ranking of ownership rates. However, Japan has a low rate of social housing stock, comprising about 5% of the total housing stock. Inadequate housing policies may increase the proportion of economically vulnerable people living in low-rent and poorly-maintained accommodation in the private rental sector. This could have adverse health effects due to financial stress and difficulties affording health care [3].

Second, Japan has among the highest proportion of older adults aged 65 and older in the world [15]. Because older people tend to spend most of their time at home and are vulnerable to barriers and problems of the home environment [16], Japanese people may be more exposed to health risks associated with their own home. Furthermore, previous studies suggest that the housing-health association can be partly explained by neighborhood environments [17,18]. A cross-national research study in China, Japan, and South Korea has reported that the association between neighborhood social environment and self-rated health is strongest in Japan [17]. Japanese culture is greatly influenced by Confucian ideals, where harmony among people is to be valued. Japanese people may tend to place emphasis on neighborly ties and therefore their health may be more affected by their neighborhood social environment. Based on these assumptions, the HT-health association may be stronger in Japan than in other countries. Therefore, we hypothesize that HT is significantly related to the health status of Japanese people, independent of other socioeconomic factors, such as education, expenditure, and occupation.

Comment 5. Methods

- Why was the 2010 survey used and not a more recent one? It was mentioned that the CSLC is an annual survey.

- Why were the researchers limited to a sample of 36,387 households when data was collected from 229,785 households (15% of the original size)?

Response:

We agree with your suggestion that we had insufficient description of the data sources in this study. Taking your comment, we have revised the Methods section as follows:

Revised parts:

Page 8-9, Line 88-110.

Data

The data source in this study is based on the 2010 Comprehensive Survey of Living Conditions (CSLC) conducted by the Ministry of Health, Labour and Welfare of Japan. The details of the 2010 CSLC are explained elsewhere [24]. Briefly, the CSLC covers households and their membership throughout Japan, and has been carried out annually for the purpose of collecting basic data for the promotion of national health and social welfare. A large-scale survey is implemented every 3 years. In other years, smaller-scale surveys are conducted with simplified questionnaires leaving out health-related questions such as SRH. In January 2018, when we received the CSLC data from the Ministry of Health, Labour and Welfare, the 2010 data was the latest information provided. For the 2010 CSLC, survey slips were distributed to all households in 5,510 stratified random sampling districts (289,363 households) on June 3, and collected from 229,785 households (response rate, 79.4%). We got permission to use this data for academic research in accordance with the Statistics Act, Article 36, and accepted an offer of anonymized data. Anonymized data were scrubbed of information that had the possibility of revealing personal identity; not only personal information such as name and birthdates, but also regional information such as prefectural names. In addition, anonymized data eliminated rare households such as single-male-parent households and families who had a large age difference between husband and wife, because including these households might lead to identiﬁcation of individuals. After rare households were excluded, individuals from the anonymized data were randomly selected. We were provided finally with anonymized data from 93,730 people in 36,387 households; the reduced size equivalent to that of smaller-scale surveys.

Comment 6. Methods

- Was more than one household member included in the sample? If so, did the authors control for potential clustering (i.e., violation of the independence assumption)?

Response:

Thank you very much for providing us with this valuable comment. We failed to control for potential clustering. To deal with this concern, we have conducted subset analyses limited to the head of a household, added the results of subset analyses in Table 2, and revised the Results section as follows:

Revised parts.

Page 18-19, Line 257-264

Given that nine out of every ten people in this study had more than one member in their household, there was a possibility that we had the violation of the independence assumption in the association between HT and SRH. To address this concern, we conducted subset analyses limited to the head of the household (n = 27,849). The results did not change signiﬁcantly with the subset analyses: in Model 3, adjusted OR (95% CI) was 1.46 (1.32-1.62) in private renters, 1.43 (1.24-1.65) in publically subsidized renters, and 1.35 (1.12-1.61) in residents in rented rooms, compared to owner-occupiers.

Comment 7. Methods

- What was the rationale for dichotomizing SRH?

- Is the dependent variable SRH (not poor SRH and good SRH was the reference category)?

-Restate the independent variable as being HT, which has five categories (owner-occupier was the reference category)

Response:

We agree with the reviewer that our explanation was inadequate on self-rated health, dependent variable, and independent variable. Therefore, in the revised draft, we have made the following amendments in the Methods section:

Revised Parts:

Page 11-12, Line 146-161.

Self-rated health (SRH)

SRH has been used as an effective indicator of overall health status not only in Japan [23,26-28] but also worldwide [10,29,30]. This study assessed SRH by a single item: “How is your health in general? Is it very good, good, fair, poor, very poor?” The OECD Health Statistics [15] has recommended this as a standard form of question about perceived health status, and showed that the rate of people who report their health as good or better is very low in Japan, about 30% in 2011 compared to about 70% on average in the OECD and about 90% in the United States, New Zealand, and Canada. One reason why Japan has a low rate of people reporting to be in good health is that Japanese people have a tendency to avoid giving a direct answer and like moderation, with the result that in responses to questionnaires, there is a marked tendency to concentrate on a mid-point [31]. Therefore, although the OECD Health Statistics has considered people rating their health to good or very good as those who are in good health, Japanese epidemiological surveys have commonly adopted the definition of good health as including the middle scale of SRH [26-28]. In this study, persons whose responses were very good, good, and fair were defined as having good SRH, and poor and very poor as having poor SRH.

Page 14, Line 205-211.

Statistical analysis 

Multiple logistic regression analysis (by the forced entry method) was carried out using ‘poor SRH’ or ‘good SRH’ as a dependent variable. Independent variables were the five types of HT (i.e., owner-occupied housing, privately rented housing, provided housing, publically subsidized housing, and rented rooms), with owner-occupied housing as the reference group. The results were shown as an odds ratio (OR) with a 95% conﬁdence interval (CI) for poor SRH.

Comment 8. Change ‘smoking habit’ to ‘smoking status’

Response:

In response to your comment, we have changed the term “smoking habit” into the term “smoking status” throughout not only the text but also the tables. 

Comment 9. Adding a flow chart showing how the cohort was created would be helpful (and would make the text in this section more concise). Suggestion to include the percentage of people excluded for each reason in addition to the frequencies.

Response:

Taking your advice, we have added a flow chart of the selection of study participants in Figure 1. Because the number of households excluded from anonymized data to prevent personal identification (i.e., rare households) was undisclosed, we could not provide the number.

Comment 10. I had to do a little background research to understand why Poisson regression was used and not logistic regression. It might be helpful to include a one sentence explanation on why Poisson regression was performed.

Response: 

Recently, some articles have discussed the advantages/disadvantages of odds ratio (OR) versus prevalence ratio (PR) and debated the “appropriate” measure of association. In the first draft, we considered PR to be more appropriate than OR due to considerable “overestimation” of the strength of the association by OR. However, because in this study the proportion of outcome is less than 15%, “overestimation” hardly matters. Therefore, it is no problem to perform logistic regression. In the revised manuscript, we have used logistic regression models, calculated the OR, and rewritten the Methods and Results sections as follows.

Revised parts:

Page 14, Line 205-211 (Methods).

Statistical analysis 

Multiple logistic regression analysis (by the forced entry method) was carried out using ‘poor SRH’ or ‘good SRH’ as a dependent variable. Independent variables were the five types of HT (i.e., owner-occupied housing, privately rented housing, provided housing, publically subsidized housing, and rented rooms), with owner-occupied housing as the reference group. The results were shown as an odds ratio (OR) with a 95% conﬁdence interval (CI) for poor SRH.

Page 18, Line 244-257 (Results).

Cross-sectional association between HT and SRH

The ORs for poor SRH associated with HT are presented in Table 2. Publically subsidized renters had a significantly higher prevalence of poor SRH. This significant relationship persisted after adjustment for demographic factors and SES factors (adjusted OR 1.33; 95% CI, 1.19-1.48 in Model 3). Persons living in provided housing were less likely to have poor SHR than owner-occupiers. However, after adjustment for age and gender, this association did not remain significant: adjusted OR (95% CI) was 1.11 (0.94-1.32) in model 1 and 1.12 (0.94-1.34) in model 3. In contrast, in the crude model, private renters and persons inhabiting rented rooms showed no association with poor SRH relative to owner-occupiers. After adjustment for age and gender, private renters and residents in rented rooms were more likely to have poor SRH than owner-occupiers. These significant associations were unchanged after adjustment for SES factors as well as demographic factors (Model 3): adjusted OR (95% CI) was 1.36 (1.26-1.47) in private renters and 1.41 (1.22-1.62) in residents in rented rooms.

Comment 11.Thanks for including the S1 Table. In the limitation section, please address any bias that may have resulted by excluding a group of people who were older than those included in the study.

Response:

In response to this comment, we have added the limitation due to anonymized data as well as selective loss of older people in the Discussion section as follows:

Revised parts:

Page 27, Line 360-365.

Third, anonymized data eliminated the rare households in order to prevent identification of individuals, and our study failed to include persons with missing data. The former ruled out social minorities and the latter resulted in selective loss of older people. That is, persons omitted from our study are vulnerable groups, and potentially have poor health and poor housing. This bias might lead to an underestimation of the association between HT and SRH in this study.

Comment 12. Results - It seems a bit odd that that those who were living in provided housing had the highest percentage married (more than 70%) and the highest percentage living alone (almost 30%). I recommend addressing this seemingly conflicting finding.

Response:

Because people living in provided housing were more likely to live apart from their family at the new workplace, they had the highest percentage married and the highest percentage living alone. In accordance with your comment, we have added an explanation of this seemingly conflicting finding as follows:

Revised parts:

Page 16, Line 233-242.

Owner-occupiers were more likely to be aged 65 years and older and have chronic medical conditions, and less likely to be current smokers and live in crowded houses; and private renters had the second highest tobacco use; Because a large number of people living in provided housing live apart from their family at the new workplace, they had the highest percentage of the married, males, and upper non-manual workers, the second highest percentage of persons living alone, and the lowest percentage of persons aged 65 or older and those who had a low level of education. Publically subsidized renters tended to have junior high school education, live in crowded homes, and report poor SRH; and those living in rented rooms were more likely to be current smokers and have a low EHE. 

Comment 13. Results - The statistical test results are missing from Table 1. It was noted below the table that chi-square and ANOVA tests were performed.

Response:

In accordance with your comment, we have added P-values based on chi-square and ANOVA tests in Table 1.

Comment 14. Discussion - This study did not include any minority groups; suggestion to drop mention of this as a vulnerable group.

Response:

As previously noted in Comment 11, our discussion of the study limitations failed to address the exclusion of minority groups from study participants. Therefore, in the revised draft, we have added the limitation due to exclusion of minority groups as well as selective loss of older people in the Discussion section with the following.

Revised parts:

Page 27, Line 360-365.

Third, anonymized data eliminated the rare households in order to prevent identification of individuals, and our study failed to include persons with missing data. The former ruled out social minorities and the latter resulted in selective loss of older people. That is, persons omitted from our study are vulnerable groups, and potentially have poor health and poor housing. This bias might lead to an underestimation of the association between HT and SRH in this study.

Comment 15. Discussion - The authors suggest that the housing environment may contribute to poor health. Is unsanitary conditions, inadequate garbage disposal and sewage treatment, poor ventilation, etc. a problem in Japan?

Response:

We agree with your suggestion that it is questionable whether unsanitary conditions are a problem in Japan these days. In the revised draft, we have revised our description of the mechanism relating to housing environment in the Discussion section as follows:

Revised parts:

Page 24-25, Line 307-325.

First, previous studies suggest that the housing environment of vulnerable groups tends to be crowded due to a large number of people living in a limited area, unsanitary living conditions due to inadequate garbage and sewage treatment, bad air with harmful chemical substances and poor ventilation, and poor hygrothermal conditions such as warmth and humidity [3,36]. Crowding and poor air quality can cause respiratory disease and communicable diseases, a polluted water supply can spread waterborne diseases, and cold homes can elevate the risk for cardiovascular diseases and poor mental health [36,37]. In Japan, because universal access to a clean water supply and an effective sanitation system has improved the level of public hygiene, unsanitary living conditions may not be an issue nowadays. However, because Japan has a high death toll during the winter months [38], excess cold in the home is an ongoing problem in Japan: People with financial difficulties may not be able to afford adequate heating, with consequent negative impact on their health. In this study, among persons living in rented rooms, household crowding was associated with SRH. Our results suggest that people living in rented rooms may be exposed to more hazards in the home environment than people living in other types of HT. For example, persons inhabiting crowded rented rooms seem likely to face the risk of intruders and excess noise. These conditions together with crowding in housing are considered as psychological hazards relating to housing, and could lead to increased risk of illness [37].

Comment 16. Discussion - In the Japanese context, why might home owners engage in healthier behaviors than other HT groups? In Japan, is social housing found in neighbourhoods with lower social capital?

Response:

Thank you very much for providing us with these valuable comments. In Japan, because there have been few studies on HT or social housing, we have been unable to find previous studies reporting the association between neighborhoods in social housing and lower level of social capital. However, to address these concerns, we have conducted literature searches, and found some studies which suggest the mechanisms relating to health behaviors and the neighborhood. Therefore, in the revised manuscript, we have revised the Discussion section as follows:

Revised parts:

Page 25-26, Line 326-343.

Second, previous studies have reported that the social housing group tend to have a lower level of physical activity and a higher prevalence of obesity than those in other housing sectors [39] and that house owners are more likely to have regular medical checkups [40] and to be non-smokers [11] than persons without house ownership; they indicate that the positive health effect of house ownership may be the result of healthy lifestyle and active healthy behaviors among owner-occupiers. These findings from previous studies agree with our results showing that, among the five types of HT, house owners have the lowest rate of current smokers. 

Third, the neighborhood environment of dwellings in the publically rented sector is often characterized by poor access to recreational facilities, shops, public transport, social support networks, and health services [6,39]. A neighborhood environment that discourages physical activity and health care may affect health behaviors and the health of residents [3]. On the other hand, research has shown that people living in a neighborhood with more sports/recreational facilities, more walkable green spaces, and more people who practice healthy lifestyles such as choosing healthy foods and doing active physical exercise are more likely to report being in good health [41-44]. The communities with a high homeownership rate are considered to have a healthy neighborhood [11], which promotes the health of house owners.

Comment 17. Discussion- From a policy perspective, how are these results useful?

Response:

In response to this comment, we have added the message for policymakers in the conclusion section as follows:

Revised parts:

Page 28, Line 367-375.

Conclusions

This study has revealed a significant association between HT and SRH in Japan, independent of SES factors and demographic characteristics. The Japanese government has adopted reduction of health disparities as one of the main target objectives for the national health promotion plan, Health Japan 21 (the second term) [47]. For a better understanding of disparities in health status among the Japanese, our findings suggest that HT is an important factor that deserves notice. From a policy perspective, policymakers need to pay more attention to HT as a social determinant that contributes to health inequality in Japan.

Comment 18. General Questions - Does the journal have a policy about the use of the term ‘subject’ versus ‘participant’?

Response: 

In response to your comment, we have changed the term “subjects” into the terms “participants” or “individuals” throughout not only the text but also the tables.

Comment 19. General Questions -- The authors used the term ‘gender’, not ‘sex’. Which word should be used in this context?

Response: 

According to the WHO, gender is defined as follows: “Gender has implications for health across the course of a person’s life in terms of norms, roles and relations. It influences a person’s risk-taking and health-seeking behaviors, exposure to health risks and vulnerability to diseases. Gender shapes everyone’s experience of health care, in terms of affordability, access and use of services and products, and interaction with healthcare providers.” We have an understanding that the use of the term “gender” is suitable for this study.

---

## [Decision Letter · Decision Letter 1]

23 Oct 2019

Association between Housing Tenure and Self-Rated Health in Japan: Findings from a Nationwide Cross-Sectional Survey

PONE-D-19-16134R1

Dear Dr. Tomioka,

We are pleased to inform you that your manuscript has been judged scientifically suitable for publication and will be formally accepted for publication once it complies with all outstanding technical requirements.

With kind regards,

Sungwoo Lim, DrPH

Academic Editor

PLOS ONE

Additional Editor Comments (optional):

Reviewers' comments:

Reviewer's Responses to Questions

**Comments to the Author**

1. If the authors have adequately addressed your comments raised in a previous round of review and you feel that this manuscript is now acceptable for publication, you may indicate that here to bypass the “Comments to the Author” section, enter your conflict of interest statement in the “Confidential to Editor” section, and submit your "Accept" recommendation.

Reviewer #1: All comments have been addressed

Reviewer #2: All comments have been addressed

2. Is the manuscript technically sound, and do the data support the conclusions?

Reviewer #1: (No Response)

Reviewer #2: Yes

3. Has the statistical analysis been performed appropriately and rigorously? 

Reviewer #1: (No Response)

Reviewer #2: Yes

4. Have the authors made all data underlying the findings in their manuscript fully available?

Reviewer #1: (No Response)

Reviewer #2: Yes

5. Is the manuscript presented in an intelligible fashion and written in standard English?

Reviewer #1: (No Response)

Reviewer #2: Yes

6. Review Comments to the Author

Reviewer #1: (No Response)

Reviewer #2: Thanks to the authors for their detailed responses to my questions/comments. I appreciate the time and effort they invested in revising the paper. I noticed two very minor corrections that needs to be made. The "b" in "Because" on page 16 line 236 should be a lowercase "b". "PR" needs to be changed to "OR" on page 21 line 268. Otherwise, this is an excellent paper and worthy of publication!

7. PLOS authors have the option to publish the peer review history of their article (what does this mean?). If published, this will include your full peer review and any attached files.

Reviewer #1: No

Reviewer #2: No

---

## [Editor Report · Acceptance letter]

7 Nov 2019

PONE-D-19-16134R1 

Association between Housing Tenure and Self-Rated Health in Japan: Findings from a Nationwide Cross-Sectional Survey 

Dear Dr. Tomioka:

I am pleased to inform you that your manuscript has been deemed suitable for publication in PLOS ONE. Congratulations! Your manuscript is now with our production department. 

With kind regards,

on behalf of

Dr. Sungwoo Lim 

Academic Editor

PLOS ONE